# The Hedgehog Signaling Pathway in Idiopathic Pulmonary Fibrosis: Resurrection Time

**DOI:** 10.3390/ijms23010171

**Published:** 2021-12-24

**Authors:** Wiwin Is Effendi, Tatsuya Nagano

**Affiliations:** 1Department of Pulmonology and Respiratory Medicine, Faculty of Medicine, Universitas Airlangga, Surabaya 60015, Indonesia; 2Department of Pulmonology and Respiratory Medicine, Universitas Airlangga Teaching Hospital, Surabaya 60015, Indonesia; 3Division of Respiratory Medicine, Department of Internal Medicine, Kobe University Graduate School of Medicine, 7-5-1 Kusunoki-cho, Chuo-ku, Kobe 650-0017, Japan; tnagano@med.kobe-u.ac.jp

**Keywords:** cell signaling, signal transduction, hedgehog pathway, chronic respiratory diseases, idiopathic pulmonary fibrosis

## Abstract

The hedgehog (Hh) pathway is a sophisticated conserved cell signaling pathway that plays an essential role in controlling cell specification and proliferation, survival factors, and tissue patterning formation during embryonic development. Hh signal activity does not entirely disappear after development and may be reactivated in adulthood within tissue-injury-associated diseases, including idiopathic pulmonary fibrosis (IPF). The dysregulation of Hh-associated activating transcription factors, genomic abnormalities, and microenvironments is a co-factor that induces the initiation and progression of IPF.

## 1. Introduction

Cell signaling is a multifactorial system that represents the knot-like schematics of the signaling cascades that are used to transfer messages from the first messenger to the receptor and decoded through the signaling intermediates of the second messengers [1,2]. Signal transduction, as an aspect of cell signaling, describes how cells interpret and react to external events [3]. Hh is one of the major signal transduction networks for intercellular communication during embryonic development and organogenesis [4] and it regulates mitogenic and morphogenic functions during organ development [5]. Nevertheless, many disease processes arise from defects or through the aberrant activation of these developmental pathways.

The dysregulation of the Hh signaling network in controlled cell growth and division induces autocrine and paracrine function distortions, leading to the development of tumorigenesis and cancer progression [6,7]. Myofibroblast-associated Hh signaling is involved in accelerated tumor growth in various cancers [8,9,10]. Moreover, Hh signaling is also responsible for the development of numerous lung diseases [11]. Recent gene expression studies and animal disease models have demonstrated that Hh signaling can induce the fibroblast to myofibroblast transition (myofibroblast differentiation) in IPF [12]. Here, we review the disturbances that may occur in the multilayered Hh signal transduction network during the development of IPF.

## 2. Hh Signal Transduction

Vertebrate genome duplication categorized Hh genes into three different types of hedgehog proteins: the Desert Hedgehog (Dhh), Indian Hedgehog (Ihh), and Sonic Hedgehog (Shh) proteins [13].

### 2.1. Element of Hh Signal Transduction

Hh proteins undergo multiple processing steps that are required for the generation and release of the active ligand from the producing cell. The core components that mediate the Hh signal response in vertebrae are two patched receptors (Ptch1, Ptch2), a key signal transducer smoothened receptor (Smo), three glioma-associated oncogene (Gli) transcription factors (Gli1, Gli2, Gli3), the suppressed fusion homolog (Sufu), and kinesine protein 7 (Kif7) [14].

#### 2.1.1. Hh Ligand

Hh genes are automatically cleaved into a 20 kDa N-terminal protein (Hh-N) and a 25 kDa C-terminal protein (Hh-C) [15]. After translation, the Hh protein is then transported to the endoplasmic reticulum for dual lipid modification. The first modification removes the C-terminal domain and attaches cholesterol to the C-terminal (the C-terminally cholesterol-modified N-terminal Hh signaling domain (HhN)), leading to the association of Hh with membranes [16]. Next, a palmitate molecule is attached to the N-terminal by Hh acyltransferase (Hhat), resulting in a fully active dual lipid-modified HhNp [17]. The dual lipid-modified HhNp is then transported to the cell surface.

#### 2.1.2. Ptch

Hh ligands start to trigger signaling in the target cells by binding a 12-pass integral membrane, the Ptch protein (complex Hh-Ptch). Vertebrates have two Ptch genes, Ptch1 and Ptch2, but Ptch1 is the primary signaling regulator [18]. Ptch1 is essential for Hh signaling and for generating stable signaling gradients due to negative feedback, the inhibition of Hh ligands and Smo, and involvement in a double-negative circuit (in which Hh inhibits Ptch and also blocks Smo). [19]. The Ptch1 protein has a sterol-sensing domain (SSD), two large extracellular loops, and a C-terminal cytoplasmic tail [20]. SSD mediates the vesicular trafficking of Ptch1 to regulate Smo activity [21].

#### 2.1.3. Smo

The G protein-coupled receptor (GPCR), the Smo protein, which is predominantly located in the membrane of intracellular endosomes, functions as a co-receptor and a positive regulator of the Hh signaling pathway. Smo consists of an amino-terminal cysteine-rich domain (CRD), three extracellular and three intracellular loops (ECL and ICL), seven transmembrane domains (TM), and an intracellular carboxyl-terminal tail that is able to undergo a range of post-translational modifications [22].

#### 2.1.4. Gli

The Gli family of latent zinc-finger proteins function as transcriptional mediators and are implicated in the activation and repression of the Hh target genes [23]. In detail, Gli1 only acts as a transcriptional activator (GliA), Gli2 is the principal Hh-regulated transcriptional activator, and Gli3 is the strongest Hh-regulated repressor (GliR) [24]. The differential activity of Gli is regulated on the level of ubiquitin-mediated proteolytic processing [25] and the subcellular localization of a nuclear localization signal (NLS) and a nuclear export signal (NES) [26].

#### 2.1.5. Sufu

Sufu is an essential intracellular negative regulator of Hh signaling and acts by binding and modulating the Gli transcription factors [27]. In the absence of signaling, Sufu inhibits the Gli transcription factors by binding Gli through a C-terminal Sufu-interacting site (SIC) that is responsible for the Sufu-mediated cytoplasmic retention of Gli1 via the N-terminal Sufu-interacting (SIN) pathway [28].

#### 2.1.6. Kif7

Kif7 is a cilia-associated protein that regulates signaling from the Smo to the Gli [29]. However, Kif7 regulates the activity of Gli through both Sufu-dependent and -independent mechanisms [30]. Additionally, Kif7 can act as both a positive and negative regulator of the Gli activity [31].

### 2.2. Hh Signaling Pathway

The Hh activates canonical (either through ligand-dependent interaction or receptor-induced signaling) and non-canonical (ligand-independent interaction) signaling pathways (Figure 1).

#### 2.2.1. Canonical Pathway

The canonical signaling pathways focus on the mechanism by which Hh regulates the Gli [32]. This pathway can be operated in both the presence and absence of Hh ligands.

In the absence of Hh ligands, Ptch1 functions to suppress any inactive Smo that is inside the cell and inhibits the migration of Smo to the membrane [33]. The mechanism through which Ptch1 inhibits Smo is not precise. It is supposed that Ptch1 is transported out of the cell as an endogenous intracellular small molecule that acts as an agonist for Smo and that does not bind to Smo [34]. Ptch1 requires extracellular Na^+^ and membrane cholesterol to regulate Smo [35]. Furthermore, Ptch1 might inhibit Smo through an indirect mechanism, possibly through changes in the distribution or concentration of a small molecule [36].

The inhibition of Smo activity is an essential step for activating this pathway in mammals [37]. The full-length Gli (GliFL) is then phosphorylated at multiple sites in the C-terminal region by protein kinase A (PKA), glycogen synthase kinase-3 (GSK3), and casein kinase 1 (CK1) [38]. Kif7 acts as a scaffolding protein for PKA, GSK3, and CK1 during the Gli phosphorylation [32]. Without Hh, Sufu restrains the GliFL protein in the cytoplasm, whereas ligand binding will proteolytically cleave Gli from Sufu [39]. The truncated GliR (Gli3) then translocates to the nucleus and binds to the Hh target gene [40].

The Ptch1 protein stops inhibiting Smo after binding the Hh ligand and limits the half-life of the ligand [41]. Smo activation induces the stabilization and release of Gli, the transducer of the significant cellular effects of canonical Hh signaling, from cytoplasmic retention [42]. The Hh–Ptch1 complex is then internalized and degraded in the lysosomes [43]. Hh signaling is subsequently activated and transmitted via a protein complex that includes Kif7 and Sufu [44]. Finally, GliFL is converted to its active form GliA (Gli1 and Gli2) and migrates to the nucleus to activate several target genes [45]. Canonical Hh signaling leads to Gli code regulation, which covers the sum of all of the positive and negative functions of all of the Gli proteins [40].

#### 2.2.2. Non-Canonical Pathway

Contrary to canonical complex signaling network resulting in activation of the Gli family of transcription factors, some Hh signaling proceeds through Gli independent activation. In detail, non-canonical Hh delivers signals via (1) Ptch1 in the presence of Hh ligand, (2) Ptch1 in the absence of Hh ligand, and (3) Smo-dependent and Gi protein modulating Ca^2+^ and actin skeleton [32,46,47,48].

After embryogenesis, the Hh pathway continues to signal to discrete populations of stem and progenitor cells within various organs in order to maintain tissue homeostasis and repair [49]. In the lungs, the Hh pathway never entirely disappears from development to adulthood, but the activation domain shifts dramatically and repurposes itself in order to maintain cellular homeostasis and organ function [50]. It seems that the Hh pathway is silenced until it is reactivated by tissue injury in order to mediate cellular regeneration and repair.

## 3. Hh Signaling in Lung Development

It is already known that the Hh signaling pathway plays a critical role as the principal regulator in the normal development of many tissues such as those in the lung. Lung morphogenesis relies on intricate interactions and the coordinated development of the endoderm layer-derived epithelial cells into the surrounding mesoderm-derived mesenchyme [51].

Embryonic lung development follows the principle of branching morphogenesis into five phases; the first four phases (embryonic, pseudo glandular, canalicular, and saccular) result in a typical branching structure that ends with alveolar sacs with a surrounding stromal scaffold and vascular structures where during the final (postnatal) alveolar phase, the terminal sacs give rise to mature alveolar ducts and alveoli [11]. The first two development stages regulate the establishment of the conducting airways, and the last three stages are responsible for vascular development, alveolar development, and reducing mesenchymal tissue, which is crucial for the formation of the thin air–blood interface that is indispensable for gas exchange [52]. Hh signaling is a crucial aspect that can be used to orchestrate a network of growth factors, transcription factors, and extracellular matrix molecules during lung embryogenesis [11].

During embryogenesis, the Shh that is secreted by the epithelial cells during the early steps of embryogenesis, act as a spatial regulator of bronchial bud formation and are essential for the mesenchymal–epithelial cross-talk that guides branching and epithelial tube elongation, as well as smooth muscle cell/myofibroblast differentiation [53]. Other elements of Hh signaling, such as Ptch1, Smo, and Gli1-3, are mainly expressed in the epithelium but are expressed weakly in the mesenchyme of the developing human lung [54]. Inhibition of the Shh pathway in mouse models causes severe lung malformations, resulting in hypoplasia and tracheal malformations and non-viable phenotypes [55]. He et al. demonstrated that Shh signaling controlled multiple morphogen signaling pathways, such as Fgf10 expression, in lung morphogenesis via heparan sulfate (HS) glycosaminoglycans [56].

In contrast with its crucial roles during embryonic development, Hh signaling has more restricted roles after birth. Postnatally, mature lung development begins with the formation of the alveolar septum (alveolarization) followed by secondary septa and microvascular maturation [57]. The Hh pathway also regulates mesenchymal proliferation and myofibroblast function during the septum alveolarization and maturation phase [58]. Overall, Hh signaling plays a vital role in lung embryogenesis, homeostasis, and regeneration via the fine cellular distribution of the Hh pathway components, which orchestrate complex cross-talk between lung cell populations, leading to proper lung development [59].

Recent studies indicate that the growth signaling pathways may be reactivated in tissue remodeling and cancer development. IPF and lung cancer share similar cellular and molecular pathological processes, including aberrant embryological pathways [60]. Hh signaling is one of the pathways that is responsible for the activation and proliferation of both the myofibroblasts in IPF and cancer-associated fibroblasts (CAF) [61].

Several studies showed the involvement of the Hh pathway during fibroblast activation and during myofibroblast transformation in biliary and liver fibrosis [62,63,64,65] as well as in kidney fibrosis [66,67]. Froidure et al. proposed that minimal aberrance in Hh signaling could induce the development and progression of pulmonary fibrosis rather than repair in a chronically injured lung [68]. However, a recent study declared that overexpression of Hh signaling in diabetic myocardial ischemia reduces cardiac fibrosis via suppressed myocardial apoptosis and improved myocardial angiogenesis [69].

## 4. Reactivation of the Hh Pathway in IPF

IPF is a chronic, progressive, irreversible disease that is characterized by the pathogenic cellular fibroblast to myofibroblast transition, the plasticity of alveolar epithelial cells, the recruitment of fibroblasts, cell–matrix interactions, and immune system activation within the alveolar wall [70]. The pathogenesis underlying pulmonary fibrosis remains elusive. Previously, persistent chronic inflammation has been shown to result in remodeling fibrosis; however, the paradigm has recently changed. Some studies have shown that fibrosis without inflammation is possible; therefore, any abnormality in the pathways that are involved in at least aberrant wound healing and/or inflammation may lead to the development of IPF [71,72].

The reactivation and deregulation of the Hh signaling pathway and its cross-talk with tumor growth factor-β (TGF-β) contribute to IPF pathogenesis by inducing myofibroblast differentiation and epithelial–mesenchymal transition (EMT) and by producing excessive amounts of extracellular matrix (ECM) [73].

The histopathological hallmark of IPF is usually interstitial pneumonia (UIP), which is characterized by variations in temporospatial heterogeneity fibrosis, the accumulation of fibroblasts (fibroblast foci), and subpleural and paraseptal honeycombing [74]. Although other cell types certainly make significant contributions, fibroblasts and alveolar epithelial and alveolar macrophages are the most crucial drivers that are involved in the progression of pulmonary fibrosis.

The alveolar epithelium consists of type I (AECI) and type II cells (AECIIs). AECIIs are capable of self-renewing and self-differentiation into AECIs [75]. Honeycombing is constructed via the hyperplasia and hypertrophy of AECIIs as well as widespread fibrotic areas [76]. Lung epithelia is susceptible to recurrent micro-injury resulting from environmental exposure. Repetitive and subclinical epithelial injury that has been superimposed onto accelerated epithelial aging, host defense abnormalities, and the dysbiosis of microbiome induce aberrant wound healing and deposition of ECM through the myofibroblasts [77]. Damage to the epithelium disrupts the basement membrane, and thus the alveolar–capillary barrier, leading to fibrin and fibronectin leakage, coagulation cascade activation, and abnormal vascular remodeling [78]. The inability of the dysfunctional epithelium to generate appropriate healing following repetitive injury is central to the pathogenesis of IPF [79].

The differentiation of airway fibroblasts to myofibroblasts (myofibroblast transdifferentiation) leads to the secretion of excessive amounts of ECM [80]. However, the primary origin of the myofibroblasts that are involved in IPF has not yet been established; hence, three cells, including resident fibroblasts, circulating bone marrow-derived progenitors, and EMT, have been proposed as potential sources of myofibroblasts [81,82]. EMT is a process by which fully differentiated AECIIs undergo a phenotypic transition into mesenchymal-like cells, such as fibroblasts and myofibroblasts, losing their epithelial functionality and characteristics [83].

Bidirectional interactions between AECIIs and fibroblasts drive the progression of IPF. AECIIs regulate the immune response to ameliorate lung injury by stabilizing host immune competence and by repairing most of the damaged epithelium [84]. Repetitive cycles of epithelial injury provoke myofibroblast differentiation and, as a response, these activated fibroblasts induce further AECII injury and death and create a vicious cycle of profibrotic epithelial cell–fibroblast interactions [85].

Monocytes and tissue-resident macrophages are innate immune cells that also play a critical role in driving tissue repair, regeneration, and fibrosis [86]. While the classic activated macrophages M1 play essential roles in wound healing after alveolar epithelial injury (proinflammation), alternatively activated macrophages M2 are required to resolve inflammatory responses in the lung or to terminate inflammatory responses in the lung (profibrotic) [87].

Recent studies have reported a dysregulation of Hh signaling in IPF. The inhibition of the Hh signaling pathway by up-regulating Sufu prevents lung fibrosis in mouse models [88]. Another study showed that the expression of Hh pathway genes and Hh-induced CXCL14 was elevated in IPF lung tissues [89]. On the contrary, the role of Gli1, but not Gli2, in bleomycin-induced fibrosis was limited [90]. Indeed, Smo-independent signaling might contribute more to hedgehog pathway activity in the pathogenesis of IPF [91].

Taken together, the successful use of Taladegib, a small-molecule inhibitor of the Hh signaling pathway that has been approved for the treatment of cancers, might inspire the use of Hh as a targeted therapy for IPF. The reactivation of Hh signaling drives pulmonary fibrosis via the dysregulation of epithelial injury, EMT, fibroblast activation and myofibroblast differentiation, and macrophage polarization (Figure 2).

### 4.1. Hh Signaling Regulates Lung Epithelial Repairment

The pathogenesis of IPF is started by a usually unknown epithelial injury. Recurrent injuries lead to epithelial apoptosis that is mediated by misfolded proteins and an unfolded protein response [92]. Epithelial injury-associated apoptosis can drive aberrant cell cross-talk and fibrogenesis regardless of the triggers [93]. Dysfunctional epithelial quality control network-associated cell crosstalk results in diverse cellular endophenotypes and molecular signatures of IPF [94]. The aberrant activation of epithelial cells may undergo transdifferentiation into EMT as a direct source of fibroblasts–myofibroblasts and as an escape from apoptosis in response to injury [95].

In normal conditions, Hh is one of the signaling cascades that regulate and maintain the balance of wound healing, chronic fibrosis, and cancer [96]. Following injury, epithelial cell Hh warn neighboring cells, generate matrix deposition and fibroblasts proliferation, preventing transudation, and communicate with other immune systems [97].

The dysfunction of Hh-associated lung epithelial repair and regeneration induce fibrogenesis. Normally, epithelial cells preserve the mesenchymal quiescence through paracrine Hh signaling, simultaneously as negative feedback to maintain epithelial quiescence [50]. Hh maintains a balance between proliferation in the acute phase of injury (downregulated as the mesenchyme proliferates) and quiescence (returns to baseline during injury resolution) [98]. Steward et al. first demonstrated that Hh pathway expression is up-regulated in lung fibrosis and plays a role in remodeling damaged lung epithelium [99]. Furthermore, the secretion of Hh signaling by the AECIIs was up-regulated after stimulation by oxidative stress [100].

Injury to the AECIIs causes the release of many soluble factors that participate in epithelial repair, including growth factors (TGF-β), cytokines, chemokines, stromal cell-derived factor-(SDF-)1, vasoconstrictor endothelin-1 (ET-1), interleukins, prostaglandins, and matrix components [92,101,102]. The concentration of SDF-1, which plays an essential role in tissue repair and remodeling, was increased in the plasma and in the lungs of humans with IPF [103,104]. Additionally, SDF-1-induced pancreatic cancer cell invasions and EMT via Hh signaling defined the molecular basis of active bidirectional communication between the SDF-1 and Hh pathways [105,106].

In response to injury, the airway epithelial cells induced TGF-β activation through αvβ6 and αvβ8 [107]. Zhang et al. demonstrated that TGF-β induced EMT and pro-fibrosis via the upregulation of YY1 expression [108]. Although the knowledge of TGF-β-Hh signaling cross talk is still limited, both pathways can directly regulate critical components of each other [109]. ET-1 is also involved in the pathogenesis of lung fibrosis via myofibroblast differentiation, angiogenesis, and EMT through interaction with TGF-β and other pro-fibrotic mediators [110,111]

In general, the various products that are created following the lung epithelial injury-related activation of the Hh signaling pathway is involved in the following fibrogenesis process, including myofibroblasts differentiation, macrophage polarization, and EMT.

### 4.2. Hh Signaling Controls Fibroblast Activation and Myofibroblast Differentiation

Fibroblast activation and myofibroblast differentiation are the central pathogenesis of pulmonary fibrosis [112,113,114]. The secretion of profibrotic cytokines and TGF-β induces the differentiation of fibroblasts to myofibroblasts that produce extensive ECM, increased tissue stiffness, and deteriorating lung function [115]. Indeed, mechanical stiffness, will activate TGF-β1, which plays a pivotal role in the ability of the ECM to influence the effect that the tissue microenvironment has on cell phenotype and the function and to promote progressive pulmonary fibrosis [113]. Cigna et al. identified that Hh signaling can activate myofibroblast differentiation [116] with and without TGF-β1 [117]. Interestingly, both the TGF-β and Hh genes could induce each other in driving myofibroblast differentiation [118]. However, compared to TGF-β, Hh stimulation was not fully adequate in myofibroblast differentiation [119].

It is known that control of Hh-associated lung fibrogenesis is varied in each Hh gene. Horn et al. demonstrated that the activation of the Hh pathway in patients with systemic sclerosis (SSc) was increased, which was characterized by the accumulation of Gli1-2, Smo, and Ptch1-2 [120]. Consequently, the inhibition of the Hh genes in SSc with GANT61 and Gli2 siRNA reduces pro-fibrotic markers and downregulates fibroblast activation [121]. Similarly, the inhibition of Gli but not Smo stimulated an antifibrotic environment and decreased lung fibrosis and lung collagen accumulation [122], whereas Hu et al. showed that Hh signaling, mainly Shh signaling, stimulated myofibroblast differentiation in a Smo- and Gli1-dependent manner and via the Gli1 activation of the α-smooth muscle actin (α-SMA) promoter [118]. In short, both the canonical and non-canonical Hh pathways regulate ECM accumulation and myofibroblast differentiation.

The Hh signaling pathway regulates myofibroblast function in parallel with other pro-fibrogenic proteins and cytokines. The cross-talk linking connective tissue growth factor (CTGF) and TGF-β induced myofibroblast differentiation and ECM production [123]. Osteopontin (OPN), a matricellular protein that is abundantly expressed during inflammation and repair, was highly up-regulated and may exert a profibrotic effect in IPF [124]. Hh up-regulated and directly promoted OPN-induced liver fibrosis [125]. Recently, Hou et al. demonstrated that Hh stimulates pulmonary fibrosis by OPN-mediated macrophage alternative activation [126].

In addition, targeting the RAS axis decreased collagen deposition, myofibroblast differentiation, and α-SMA expression via the inhibition of the Hh genes that are involved in silicosis [127]. Next, Cao and his colleagues found that the Shh pathway regulates myofibroblastic activation and pulmonary fibrosis via cross-talk with Wnt10a signaling [128]. A recent study on ligamentum flavum (LF) fibrosis affirmed that Wnt1-inducible signaling pathway protein 1 (WISP-1)–Hh cross-talk was a novel profibrotic pivot [129].

A hypoxic microenvironment drives fibrosis progression. Wang et al. showed that hypoxia significantly enhanced the expression of Hh signaling in pulmonary vascular smooth cell proliferation [130]. Next, hypoxia-induced CTGF-activated Hh signaling in α-SMA and collagen accumulation [131]. Altogether, it is supposed that Hh signaling dysregulation combined with various growth factors, genetics, and the microenvironment could determine lung fibrogenesis.

### 4.3. Hh Signaling Regulates EMT

Even though the specific roles of EMT in IPF have been widely hypothesized; its precise mechanisms are not entirely understood. Kalluri and Weinberg classified three classes of EMT. Type 1 EMT regulates embryogenesis and organ development. Next, type 2 EMT determines routine wound healing, tissue regeneration, and inflammation-associated organ fibrosis. On the other hand, type 3 EMT has been linked with the epithelial cells, which underwent genetic and epigenetic changes and then transformed into cancer cells [132].

The Hh pathway regulates the EMT mechanism of lung fibrosis through cross-talk with various EMT-activating transcriptional factors, such as zinc finger E-box–binding homeobox (ZEB) and Snail 1/2, and by responding to signals from the microenvironment [133]. Furthermore, the EMT of the AECIIs indirectly promotes a pro-fibrotic microenvironment through the dysregulation of paracrine signaling between epithelial and mesenchymal cells rather than by directly affecting mesenchymal population [134,135]. Hh signaling was involved in the ZEB1-mediated EMT of the AECII augments that were involved in local myofibroblast differentiation via paracrine signal tissue plasminogen activator (tPA) [136]. The inhibition of Hh signaling could prevent house dust mite-promoted EMT, which is characterized by the downregulation of mesenchymal markers, fibroblast-specific protein 1 (FSP1), and type I collagen and the increased expression of the adherens junction protein E-cadherin [137].

EMT might also be induced by various growth factors. Blocking the Hh pathway in the primary cilium abrogated the TGF-β1-induced mesenchymal differentiation of the AECIIs [138]. Furthermore, a novel study proved that mesenchymal stem cells (MSC) convert the EMT process in LPS-induced lung injury by blocking nucleus factor κβ (NF-κβ) and the Hh pathways [139]. A new study by Mammoto et al. showed that hypoxia-induced endothelial Twist1, which stimulated the accumulation of α-SMA [140]. Even the precise mechanism of Hh pathway-induced EMT in IPF pathogenesis is still limited; recent evidence supports that the cross-talk between Hh genes with other developmental pathways, growth factors, and microenvironments induce fibrogenesis.

### 4.4. Hh Signaling Regulates Macrophage Activation and Polarization

The macrophage balance in the absence of injury, in the early phases of injury, and in the fibrotic phase is different. Misharin et al. revealed that the monocyte-derived alveolar macrophages (Mo-AMs) that persevere in the lung after injury resolution express higher proinflammatory and profibrotic genes than tissue-resident alveolar macrophages (TR-AMs) [141]. However, by stimulating complex microenvironment factors, both Mo-AMs and TR-AMs can be polarized into macrophage M1 or M2 phenotypes in IPF [142].

Hh-associated macrophage function promotes fibrogenesis. Several pieces of evidence have demonstrated Hh genes were expressed in human monocytes and macrophages; therefore, Hhs act as potent chemoattractants [143,144]. Additionally, Pereira and his colleagues showed that Hh signaling-induced M2 activation leads to hepatic fibrosis and angiogenesis [145]. The inhibition of Hh signaling in breast cancer altered macrophage polarization by setting the macrophages on a path that caused them to revert to M1 macrophages [146]. It has been stated that OPN plays a role in IPF pathogenesis. Shh promoted an OPN-dependent mechanism for M2 polarization by activating the JAK2/STAT3 signaling pathway [126].

## 5. Targeting the Hh Signaling Pathway as Therapy for Pulmonary Fibrosis

Since Hh resembles the Achilles’ heel in IPF, targeting the Hh signaling pathway and Hh-related tumor microenvironments can help achieve better therapy outcomes. Ligustrazine reduced the expression of the profibrotic factor, suppressed total collagen production, and ameliorated oxidative stress, resulting in attenuating paraquat (PQ)-induced lung fibrosis [147]. Recently, microcystin–leucine–arginine (MC-LR)-induced liver fibrosis was abrogated via downregulated Gli1/2 gene expression [148]. Zhang et al. conducted in vivo and in vitro studies that confirmed that Astilbin attenuated lung fibrosis by suppressing the Hh pathway [149].

A variety of Smo antagonists that target Hh signaling have been developed for cancer and fibrosis diseases. Vismodegib is the first oral Hh inhibitor that has been approved to treat patients with locally advanced or metastatic basal cell carcinoma [150]. However, a phase 1b study determining the appropriateness of using vismodegib for IPF treatment was discontinued due to safety issues [91]. Another Smo antagonist, cyclopamine, reduced the migration and myofibroblast differentiation of human dermal fibroblasts [151]. Taladegib is a potent, synthetic, small-molecule inhibitor of Smo, which suppresses Shh signaling [152]. Currently, a phase 2 study testing Taladegib as a monotherapy in patients with IPF is still ongoing [153].

Furthermore, targeting the Gli transcription factor in the nucleus could be a therapeutic option. The administration of the Gli inhibitor GANT61 reduced lung fibrosis and lung collagen accumulation in a mouse model [154]. A study comparing SSc-ILD patients and normal subjects determined that Pirfenidone has antifibrotic effects on fibroblasts via the inhibited phosphorylation of GSK-3β interference via the Hh pathway [155].

## 6. Conclusions

The aberrant activation of Hh signaling is one of the core pathways that is involved in the development of pulmonary fibrosis. Recent evidence has shown high Hh signaling expression in the epithelial cells, fibroblasts, myofibroblasts, and macrophages, as well as in other IPF cells. When uncontrolled, the activation of Hh signaling is responsible for the development growth factors and other various products following a tissue-associated epithelial injury, the progressive accumulation of ECM by the myofibroblasts, crosstalk between the fibroblasts and EMT, and M2 macrophage polarization.

Even though IPF progression could potentially be reversed through the targeting of Hh genes, the detailed mechanism of the Hh pathway in lung fibrosis and other organ fibrosis remains to be investigated in more detail. Hh signaling regulates mesenchymal and epithelial quiescence/proliferation during normal repairs. In IPF patients, the presence of a repetitive injury induces chronic epithelial proliferation more than mesenchymal quiescence. Instead, it seems different in the ischemic heart; Hh signaling reduces fibrosis. These discrepancies might be because Hh is acutely and not chronically overexpressed in the ischemic heart.

Furthermore, a concept “apoptosis paradox” in IPF, where epithelial apoptosis can stop collagen deposition, but apoptosis resistance in myofibroblasts leads to increased fibrosis, may correlate with Hh non-canonical signaling. Therefore, further research is still needed to discover the detailed Hh signaling pathway-related inflammation and abnormal repairment mechanism in IPF.

## Figures and Tables

**Figure 1 ijms-23-00171-f001:**
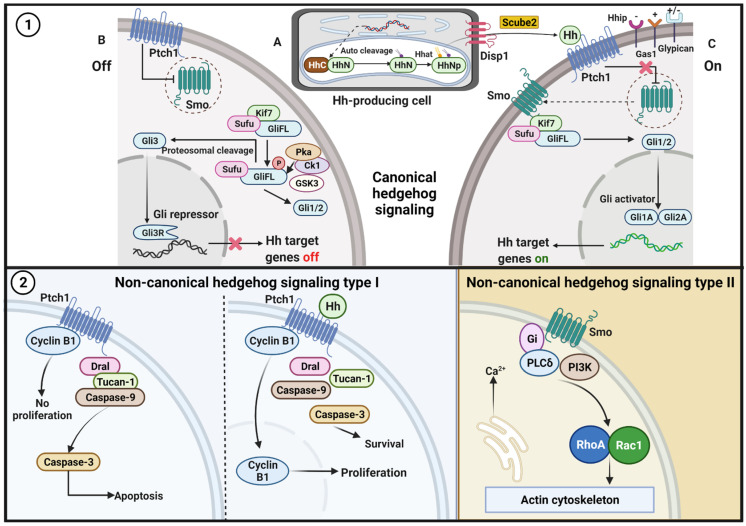
Hh pathway. (1) Canonical signaling. **A.** The production and secretion of Hh ligands/proteins. After translation, the precursor of the Hh protein is transported to the endoplasmic reticulum (ER) for autoclaved and dual-lipid modification. The first modification replaces the C-terminal domain from Hh-N with cholesterol at the C terminus; then, a palmitate molecule is attached to the N-terminal Hh-N by the Hh acyltransferase (Hhat). Disp1 on the cell surface and Scube2 regulates Hh-N secretion and distribution into the extracellular space of the producing cells. **B.** In the absence of the Hh ligands, Ptch functions to suppress any inactive Smo that is inside the cell and inhibits the migration of Smo to the membrane. Sufu restrains the GliFL protein in the cytoplasm, and GliFL is then phosphorylated at multiple sites in the C-terminal region by PKA, GSK3, and CK1. Next, the truncated GliR (Gli3) translocates to the nucleus and binds to the Hh target gene (target gene off). **C.** Ptch1 binds to the Hh ligand and releases the Smo that has been inhibited by Ptch. Active Smo induces the release of GliFL from cytoplasmic retention. The Hh–Ptch1 complex is then internalized and is degraded in the lysosomes. In the end, GliFL is converted to its active form GliA (Gli1 and Gli2) and migrates to the nucleus to activate several target genes (target gene on). Coreceptors for the Hh ligand pathway activate positive regulators (Gas1), negative regulators (Hhip), and dual functions (Glypican). (2) Non-canonical signaling. Type I, in the absence of Hh ligand, Ptch recruits complex proteins Dral, Tucan-1, and caspase-9 that followed by caspase-3 activation, which further amplifies cell apoptosis. The binding of Hh ligand to Ptch disorganizes the interaction of Cyclin B1 and the proapoptotic complex, leading to increased proliferation and survival. Type II, activation of Smo leads to dissociation of Gi, activation of PI3K and RhoA and Rac1, which then modulate the actin cytoskeleton and induce elevation of intracellular calcium.

**Figure 2 ijms-23-00171-f002:**
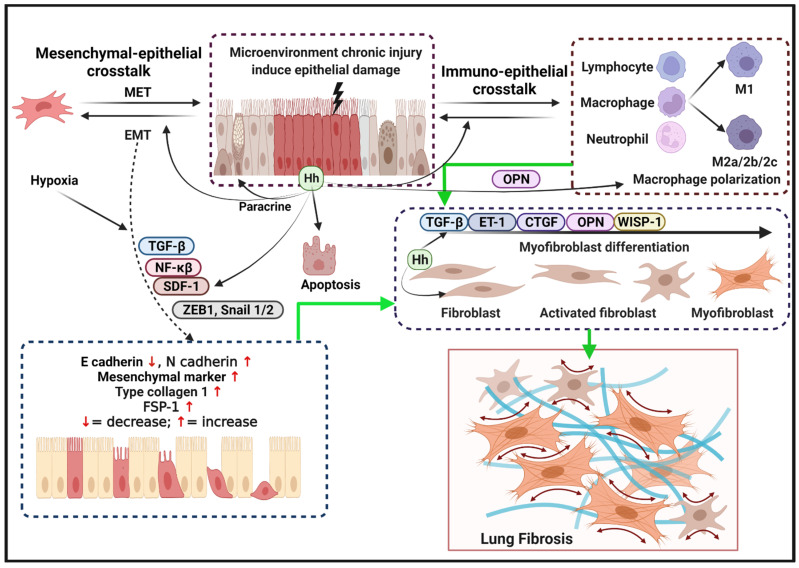
Model of uncontrolled reactivation of Hh signaling in the pathogenesis of IPF. Microenvironment factors provoke repetitive epithelial injury followed by secretion of the Hh signaling pathway-regulated various products, such as TGF-β, SDF-1, Snail 1/2, OPN, ZEB-1, and NF-κβ. Epithelial cell Hh also warns neighboring cells via paracrine signals, induces apoptosis, and initiates crosstalk with immune and mesenchymal cells. Growth factor and microenvironment-induced epithelial and mesenchymal crosstalk promote the formation of EMT characterized by increased N-cadherin, mesenchymal markers, FSP1, and type I collagen. Hh signaling as one of activating transcriptional factors AECIIs regulates the immune response to ameliorate lung injury by undergoing EMT mechanism, promoting macrophage M2-associated inflammatory components, and fibroblasts recruitment to the injured site. Furthermore, Hh signaling pathway regulates myofibroblast differentiation and ECM production in parallel with other pro-fibrogenic proteins and cytokines, including CTGF, TGF-β, α-SMA, ET-1, OPN, and WISP-1. In conjunction with other pro-fibrogenic factors and cytokines, the Hh pathway regulates the accumulation of ECM-associated myofibroblast, collagen synthesis, and lung architecture is replaced with scar tissue fibrosis.

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
