# Peer review of "The Hedgehog Signaling Pathway in Idiopathic Pulmonary Fibrosis: Resurrection Time"

_ijms, 2021, doi:10.3390/ijms23010171_

Round 1

Reviewer 1 Report

The Hedgehog Signaling Pathway is one of the interesting cell signaling pathways. The author explained the Hh-associated signaling pathways in Idiopathic Pulmonary Fibrosis (IPF).

I have some major and minor issues in this review paper.

 Major Issues

In figure 1, the author explained the canonical activation of Hh signaling pathway. I recommend that strengthening and clarifying both canonical and non-canonical signaling pathways in the same figure. The studies show that IPF is associated with an increased risk of lung cancer. There is a direct link between Hh-associated signaling pathways and IPF. I expected if this is a major theme in the paper, it should be discussed in more detail in the paper.

It seems to me that Figure 2 is not vital and does not explain any important messages or major pathways presented in the paper. I would suggest that please give more details that are relevant to the topics and it could be helpful for the readers. For instance, give the details about what are signaling molecules from damaged respiratory epithelial cells act on EMT, macrophage, and myofibroblast at the receptor level. Figure 2 needs  lot of changes.

The conclusion is very short. The author needs to state the details in the conclusion including some open questions and future direction (or missing part) of research in this field.

Minor Issues

In line 285, section 4 starts but in the following subdivision are as follows 3.1, 3.2, etc. It should be 4.1, 4.2, etc.

Author Response

Review report (reviewer 1)

The Hedgehog Signaling Pathway is one of the interesting cell signaling pathways. The author explained the Hh-associated signaling pathways in Idiopathic Pulmonary Fibrosis (IPF).

I have some major and minor issues in this review paper.

 Major Issues

In figure 1, the author explained the canonical activation of Hh signaling pathway. I recommend that strengthening and clarifying both canonical and non-canonical signaling pathways in the same figure. The studies show that IPF is associated with an increased risk of lung cancer. There is a direct link between Hh-associated signaling pathways and IPF. I expected if this is a major theme in the paper, it should be discussed in more detail in the paper.

It seems to me that Figure 2 is not vital and does not explain any important messages or major pathways presented in the paper. I would suggest that please give more details that are relevant to the topics and it could be helpful for the readers. For instance, give the details about what are signaling molecules from damaged respiratory epithelial cells act on EMT, macrophage, and myofibroblast at the receptor level. Figure 2 needs  lot of changes.

The conclusion is very short. The author needs to state the details in the conclusion including some open questions and future direction (or missing part) of research in this field.

Minor Issues

In line 285, section 4 starts but in the following subdivision are as follows 3.1, 3.2, etc. It should be 4.1, 4.2, etc.

Author`s response

Thank you for your revisions and suggestion

  1. For figure 1, we have already revised this figure by adding non-canonical Hedgehog signaling pathways. Also, we added some information about non-canonical Hedgehog signaling pathways (line 150-154; page 4)
  2. In order to strengthen the role of Hh in the pathogenesis of IPF, we add subsection (Hh signaling regulates lung epithelial repairment) in section 4 (Reactivation of the Hh pathway in IPF) to discuss more various products of lung epithelial injury-related activation of the Hh signaling pathway. These growth factors, chemokines, cytokines, and other proteins will be involved in the following fibrogenesis process, including myofibroblasts differentiation, macrophage polarization, and EMT.) (line 271-328; pages 6-8)
  3. As requested by the reviewer, we have included 1 new figures (figure 2) to depict abnormal reactivation of Hh signaling in the pathogenesis of IPF starting from the epithelial product associated with Hh signaling after injury to the development of fibrosis tissue.
  4. We agree with the Reviewer that conclusions should inform several questions/enigmatical issues related with Hh signaling and future direction of research. (line 440-458; page 10)
  5. We have already revised section 4 with the correct one.

Reviewer 2 Report

The Hedgehog Signaling Pathway in Idiopathic Pulmonary Fibrosis: Resurrection Time by Wiwin Is Effendi and Tatsuya Nagano is an extensive and comprehensive review regarding the role of Hedgehog signaling pathway in Idiopathic Pulmonary Fibrosis. The authors highlight the impairment of Hedgehog Signaling associated transcription factors, regulation of epithelial mesenchymal transition, fibroblast and macrophage activation and differentiation and its potential role as a therapeutic target. The manuscript is relevant to the field and well structured, however I have two major concerns which should be addressed:

1) The introductory section regarding function of Hedgehog signaling pathway is way too long and exceeds the aim of this review, I suggest a comprehensive handling of the section with a sensible shortening.

2) English syntax and style should be improved and undergo a major handling from a proofreader with full professional proficiency in academic English.

As a minor point I would suggest implementing section 4 (Reactivation of Hh pathway in IPF) with a short comment on the relation of aging and developmental drift as in Selman M, López-Otín C, Pardo A. Age-driven developmental drift in the pathogenesis of idiopathic pulmonary fibrosis. Eur Respir J. 2016;48(2):538-552. doi:10.1183/13993003.00398-2016. I would also integrate with the link of SHH  with lung cancer as in Samarelli AV, Masciale V, Aramini B, et al. Molecular Mechanisms and Cellular Contribution from Lung Fibrosis to Lung Cancer Development. Int J Mol Sci. 2021;22(22):12179. Published 2021 Nov 10. doi:10.3390/ijms222212179 and Ballester B, Milara J, Cortijo J. Idiopathic Pulmonary Fibrosis and Lung Cancer: Mechanisms and Molecular Targets. International Journal of Molecular Sciences. 2019; 20(3):593. https://doi.org/10.3390/ijms20030593

Author Response

Review report (reviewer 2)

The Hedgehog Signaling Pathway in Idiopathic Pulmonary Fibrosis: Resurrection Time by Wiwin Is Effendi and Tatsuya Nagano is an extensive and comprehensive review regarding the role of Hedgehog signaling pathway in Idiopathic Pulmonary Fibrosis. The authors highlight the impairment of Hedgehog Signaling associated transcription factors, regulation of epithelial mesenchymal transition, fibroblast and macrophage activation and differentiation and its potential role as a therapeutic target. The manuscript is relevant to the field and well structured, however I have two major concerns which should be addressed:

1) The introductory section regarding function of Hedgehog signaling pathway is way too long and exceeds the aim of this review, I suggest a comprehensive handling of the section with a sensible shortening.

2) English syntax and style should be improved and undergo a major handling from a proofreader with full professional proficiency in academic English.

As a minor point I would suggest implementing section 4 (Reactivation of Hh pathway in IPF) with a short comment on the relation of aging and developmental drift as in Selman M, López-Otín C, Pardo A. Age-driven developmental drift in the pathogenesis of idiopathic pulmonary fibrosis. Eur Respir J. 2016;48(2):538-552. doi:10.1183/13993003.00398-2016. I would also integrate with the link of SHH  with lung cancer as in Samarelli AV, Masciale V, Aramini B, et al. Molecular Mechanisms and Cellular Contribution from Lung Fibrosis to Lung Cancer Development. Int J Mol Sci. 2021;22(22):12179. Published 2021 Nov 10. doi:10.3390/ijms222212179 and Ballester B, Milara J, Cortijo J. Idiopathic Pulmonary Fibrosis and Lung Cancer: Mechanisms and Molecular Targets. International Journal of Molecular Sciences. 2019; 20(3):593. https://doi.org/10.3390/ijms20030593

Author`s response

Thank you for your revisions and suggestion

  1. We agree with the Reviewer that the explanation of the function of the Hedgehog signaling pathway is way too long and consider making this section more comprehensive but straightforward.
  2. As requested by the Reviewer, we submitted to MDPI for professional English editing.
  3. We add new information based on reviewer`s references
    1. Selman et al. (line 221-224; page 5)
    2. Aramini et al. (line 200-203; page 5)
    3. Ballester et al. (line 208-210; page 5)

Round 2

Reviewer 2 Report

The manuscript has been thoroughly improved.